# Starchy Vegetables and Metabolic Syndrome in Costa Rica

**DOI:** 10.3390/nu13051639

**Published:** 2021-05-13

**Authors:** Zhongyao Li, Dongqing Wang, Edward A. Ruiz-Narváez, Karen E. Peterson, Hannia Campos, Ana Baylin

**Affiliations:** 1Department of Nutritional Sciences, School of Public Health, University of Michigan, Ann Arbor, MI 48109, USA; zhongyli@umich.edu (Z.L.); eruiznar@umich.edu (E.A.R.-N.); karenep@umich.edu (K.E.P.); 2Department of Epidemiology, School of Public Health, University of Michigan, Ann Arbor, MI 48109, USA; dqwang@hsph.harvard.edu; 3Department of Environmental Health Sciences, School of Public Health, University of Michigan, Ann Arbor, MI 48109, USA; 4Centro de Investigacion e Innovacion en Nutricion Traslacional y Salud, Universidad Hispanoamericana, San Hose 40101, Costa Rica; hcampos@hsph.harvard.edu; 5Department of Nutrition, Harvard T. H. Chan School of Public Health, Boston, MA 02115, USA

**Keywords:** Costa Rican adults, metabolic syndrome (MetS), total starchy vegetables, unhealthy starchy vegetables, healthy starchy vegetables, potatoes, purple sweet potatoes

## Abstract

Only a few studies primarily examined the associations between starchy vegetables (other than potatoes) and metabolic syndrome (MetS). We aimed to evaluate the association between starchy vegetables consumption and MetS in a population-based sample of Costa Rican adults. We hypothesized that a higher overall intake of starchy vegetables would not be associated with higher MetS prevalence. In this cross-sectional study, log-binomial regression models were used to estimate prevalence ratios (PRs) of MetS across quintiles of total, unhealthy, healthy starchy vegetables, and individual starchy vegetables (potatoes, purple sweet potatoes, etc.), among 1881 Costa Rican adults. Least square means and 95% confidence intervals (CIs) from linear regression models were estimated for each MetS component by categories of starchy vegetable variables. Higher intakes of starchy vegetables were associated with a higher prevalence of MetS in crude models, but no significant trends were observed after adjusting for confounders. A significant inverse association was observed between total starchy and healthy starchy vegetables consumption and fasting blood glucose. In this population, starchy vegetables might be part of a healthy dietary pattern.

## 1. Introduction

Metabolic syndrome (MetS), defined as a group of risk factors or metabolic abnormalities including glucose intolerance, insulin resistance, central obesity, dyslipidemia, and hypertension [1], is associated with increased risks of diabetes and cardiovascular disease [2]. The prevalence of MetS increased in conjunction with the increasing prevalence of obesity over the past two decades [3]. Globally, the prevalence of MetS ranges from 8% (India) to 24% (United States) in males, and 7% (France) to 46% (India) in females [4]. In Central America, Costa Rica has the highest prevalence of MetS (35%), as compared to other countries [5]. MetS is becoming a major public health issue [6], and strategies for preventing this global epidemic syndrome are urgently needed.

Emerging studies investigated the associations of different dietary patterns with the prevalence of MetS, but few specifically examined the potential cardiometabolic impacts of starchy vegetables. The literature on this topic shows that not all starchy vegetables are equal with regards to health outcomes. For example, a higher intake of potatoes was shown to increase the risk of developing hypertension in both adult males and females [7]. However, corn was shown to assist with absorption and insulin regulation, as a result of their phytochemical content [8]. Purple sweet potatoes was shown to decrease blood glucose levels in a diabetic rats study due to the antioxidant characteristics of their flavonoid content [9]. Moreover, the molecular characteristics of amylose-to-amylopectin ratio of sweet potatoes [10] might also contribute to variations on starch contents, with respect to metabolic responses, as compared to other starchy vegetables. Most studies on starchy vegetables were conducted in developed countries, where potatoes tend to be the main starchy vegetable in the diet and are highly correlated with a Western dietary pattern that is rich in red meat, and high in saturated fat and refined flours. Costa Rica is an ideal setting to conduct a study on starchy vegetables, given that not only potatoes, but also corn, plantains, purple sweet potatoes, and cassava are commonly consumed and contribute to their dietary patterns. 

The aim of this study was to evaluate the association of starchy vegetables consumption with MetS prevalence, along with each individual MetS components in a population-based sample of Costa Rican adults. Given that starchy vegetables are diverse in nutrition composition and are associated with health outcomes, and considering that starchy vegetables in Costa Rica are not always part of a Western dietary pattern, we hypothesized that a higher overall intake of starchy vegetables would not be associated with higher MetS prevalence in Costa Rican adults, while a higher intake of potatoes could still be associated with a higher MetS prevalence. 

## 2. Materials and Methods

### 2.1. Study Population 

All subjects of this cross-sectional analysis were population-based controls from a previously conducted case-control study of nonfatal myocardial infarction (MI) in the central valley area of Costa Rica, between 1994 and 2004 [11]. The detailed case-control study methods were described in a previously published article [11]. The controls were randomly selected using information from the National Census and Statistics Bureau of Costa Rica and matched to cases of nonfatal acute MI, by age (±5 years), sex, and area of residence (county) [11]. Thus, the controls included in the study were representative of the entire Costa Rican population within the matching strata (i.e., age, sex, and area of residence). A high participation rate of controls (88%) was reported at the end of the case-control study. All subjects provided written informed consent, in order to participate in the study. 

### 2.2. Data Collection 

The data collection methods were described in detail in previously published articles [11,12]. In short, demographic data, socioeconomic data, medical histories, anthropometric measurements (height, weight, waist circumference, hip circumference), fasting lipid profiles (plasma cholesterol, triglycerides, HDL cholesterol levels), blood pressure, and fasting blood glucose levels were collected by trained personnel during home visits. Fasting lipid profile measurements were compliant with the National Heart Lung and Blood Institute Guidelines (NHLBI), while blood pressure measurements were taken, following the Dietary Approaches to Stop Hypertension (DASH) standardized protocol, as well as NHLBI [13].

### 2.3. Dietary Assessment 

A semi-quantitative Food Frequency Questionnaire (FFQ) explicitly developed and validated for the Costa Rican population [14] was used to assess dietary intake. Subjects were asked to report the usual frequency of the standard portion size of particular food intakes during the previous year. Starchy vegetables listed in the FFQ included French fries, baked/boiled/mashed potatoes, potato chips, purple sweet potatoes, plantains, corn, and cassava. Subjects were asked to categorize the frequencies of intake of starchy food items as <1 time/month or never, 1–3 times/month, 1 time/week, 2–4 times/week, 5–6 times/week, 1 time/day, 2–3 times/day, 4–5 times/day, and >6 times/day. A fixed portion size was included for each food item in the FFQ. Portion size was estimated from seven 24-h food recalls in a validation study, to estimate the average portion size in the Costa Rican population [14]. The average portion size for starchy vegetables were estimated as follows—61 g for plantains, 26 g for corn, 81 g for baked/boiled/mashed potatoes, 58 g for purple sweet potatoes, 94 g for cassava, 81 g for French fries, and 28.4 g for potato chips [14].

### 2.4. Metabolic Syndrome (MetS)

MetS was defined by the National Cholesterol Education Program (NCEP) Adult Treatment Panel III (ATP III) as the subject meeting three or more of the following criteria—abdominal obesity (waist circumference ≥102 cm in men or 88 cm in women); hypertriglyceridemia (triglycerides ≥150 mg/dL or on drug treatment for elevated triglycerides); low HDL cholesterol (high-density lipoprotein cholesterol <40 mg/dL in men or 50 mg/dL in women or on drug treatment for reduced HDL-C); high blood pressure (blood pressure ≥130 mm Hg systolic blood pressure or ≥85 mm Hg diastolic blood pressure or on antihypertensive drug treatment); and high glucose (fasting glucose ≥100 mg/dL or on drug treatment for elevated glucose) [13].

### 2.5. Data Analysis 

Eligible participants from the Costa Rica MI case-control study [11] consisted of 2274 subjects. We excluded the following subjects—(1) if it was not possible to classify the presence or absence of MetS due to missing data on more than two MetS components (*n* = 29), (2) subjects who had missing data on major potential confounders (current smoking status, history of hypertension, and BMI; *n* = 21), (3) subjects who had caloric intakes greater than 5000 kcals or less than 500 kcals (*n* = 19), and (4) subjects who had a history of diabetes (as those with diabetes were likely to change their starchy vegetable consumption; n = 324). After the exclusion, 1881 subjects remained in the analytic sample for this report. 

Food intake frequencies reported on the FFQ were first recoded into semi-continuous variables representing the nine possible responses of standard servings per day showing on the FFQ (from 0, never consumed, to 6, 6 times/day). Then, they were multiplied by the standard portion sizes to generate continuous weighted dietary variables (in grams). The weighted dietary variables were further adjusted for total energy intake by using the residual method, and were used in the statistical models [15].

Baseline sociodemographic characteristic, dietary variables, and each MetS components were investigated by the MetS status. To test the differences, chi-square tests for binary variables were used and two-sample *t*-tests were used for normally distributed variables, whereas Wilcoxon tests were used for the non-normally distributed variables. Starchy vegetables were further categorized into three main exposure variables: (1) unhealthy starchy vegetables—potatoes including French fries, baked/boiled/mashed potatoes and potato chips; (2) healthy starchy vegetables—plantain, purple sweet potatoes, corn and cassava; and (3) total starchy vegetables—the sum of unhealthy and healthy starchy vegetables. Prevalence ratios (PRs) and 95% confidence intervals (CIs) for MetS associated with the three main exposure variables and each individual starchy vegetable (all energy-adjusted weighted dietary variables), were estimated by multivariable log-binomial regression models. Confounders were selected based on prior knowledge and the association of the confounder with the dietary exposures were evaluated among people free of MetS. The first model was adjusted for age (continuous), sex, income (continuous), current smoking status (never, former or current <10, 10–20, or 20 cigarettes/day), and urban residence (urban or rural). The final model was additionally adjusted for history of hypertension (yes or no), alcohol (never, former, or current tertiles), and total energy expenditure (continuous). In all models, the indicator method was used to address the missing income data. Other potential confounders such as history of high cholesterol (yes or no) and total fat intake (continuous) were examined as covariates but these were not retained in the final model as they did not distinctly impact the effect estimates. In order to deal with model convergence issue, a modified Poisson regression with robust variance estimation was used [16]. We used the medians of exposure variables by quintiles as continuous variables to perform linear trend tests. The association between starchy vegetable intake and continuous measures of each MetS components (waist circumference, triglycerides, HDL cholesterol, blood pressure, and fasting blood glucose) was examined with linear regression. Least square means and 95% CIs were estimated for each of the MetS components, by categories of starchy variables. The multiplicative interaction between exposure variables and potential effect modifiers (age, sex, and BMI) was tested by including the interaction terms in the final fully adjusted log-binomial models.

Theoretical substitutions were assessed with substitution models. The potential association of substituting 50 g of non-starchy vegetables for 50 g of starchy vegetables with MetS, were investigated by using servings (standardized to 50 g) as continuous variables in the same final fully adjusted models [17]. The potential impact of substituting 50 g of unhealthy starchy vegetables for 50 g of healthy starchy vegetables was also estimated. Prevalence ratios and 95% CI were calculated to estimate the associations between the substitution of interests and MetS. Non-starchy vegetables included the sum of energy adjusted weighted servings of tomatoes, cucumber, zucchini, string beans, broccoli, cabbage, cauliflower, eggplant, spinach, kale, iceberg lettuce, romaine lettuce, celery, green peppers, onions, and cilantro.

In order to examine the robustness of the results, sensitivity analyses were performed. First, waist circumference was adjusted for age and BMI, by using the residual method. Second, the analysis was repeated after excluding subjects with history of hypertension. Third, least square means for systolic blood pressure, diastolic blood pressure, and fasting blood glucose were calculated after excluding subjects who reported taking hypertension medication (calcium channel antagonists, beta blocker, diuretics, or other antihypertensives) or diabetes medication (insulin or oral hypoglycemic agents). Fourth, the analysis was repeated on the main exposure variables after excluding people who reported that they had changed their long-term intake of fruits and vegetables in the last 10 years, previous to the data collection. Finally, the substitution analysis was repeated in the same fully adjusted model after excluding subjects with history of hypertension or history of hypercholesterolemia. All data analyses were performed by using the SAS software, version 9.4 (SAS Institute Inc., Cary, NC, USA).

## 3. Results

The majority of general characteristics and dietary factors were significantly different between subjects without MetS (*n* = 1123) and subjects with MetS (*n* = 758), except for urban residence, income, total fat as a percentage of total energy, and protein as a percentage of total energy (Table 1). Subjects with MetS were more likely to be older, female, and to have a history of hypertension, whereas those without MetS were more physically active and more likely to be current smokers. Characteristics of the study population without MetS across quintiles of total starchy vegetables were used to guide our selection of potential confounders (Appendix A).

The crude models showed significant higher prevalence of MetS across the quintiles of starchy vegetables. However, after adjusting for confounders, no significant trends were observed (Table 2). Of note, although not significant, the fully adjusted model for healthy starchy vegetables showed prevalence ratios in the protective range. Additional analysis of the association of foods with MetS showed that this might be driven by purple sweet potatoes. Despite not reaching significance, the PR for the highest quintile for purple sweet potato intake was 0.88 (95% CI: 0.75, 1.04; P for trend: 0.2853). Unhealthy starchy vegetables, on the other hand, did not show a significant association with MetS in the fully adjusted model (P for trend: 0.9632). Neither baked/boiled/mashed potatoes, potato chips, or French fries were associated with a higher prevalence of MetS in the fully adjusted model (PR_baked/boiled/mashed potatoes 5th quintile_ vs. _1st_: 0.98, P for trend: 0.7387; PR_potato chips 5th quintile vs. 1st._: 0.95, P for trend: 0.7261; PR_French fries 5th quintile vs. 1st_: 1.02, P for trend: 0.8926). For total starchy vegetables and healthy starchy vegetables, a higher consumption was associated with lower fasting blood glucose in the fully adjusted models (P_total starchy_ for trend: 0.0040, P_healthy starchy_ for trend: 0.0236, respectively). The difference in glucose concentrations between the lowest and highest quintile was 5.1 mg/dL for the total starchy vegetables and 3.5 mg/dL for the healthy starchy vegetables (Table 3). 

Substituting non-starchy vegetables for the three types of starchy vegetables were not found to be associated with a significantly different prevalence of MetS (PR_total starchy vs. total non-starchy_: 1.06, 95% CI: 0.99, 1.15; PR_unhealthy starchy vs. total non-starchy_: 1.06, 95% CI: 0.98, 1.15; PR_healthy starchy vs. total non-starchy_: 1.06, 95% CI: 0.97, 1.17). Similarly, substituting healthy starchy vegetables for unhealthy starchy vegetables was not associated with a significantly different prevalence of MetS (PR_unhealthy starchy vs. healthy starchy_: 1.01, 95% CI: 0.91, 1.13).

In the sensitivity analysis, the effect estimates remained consistent after excluding 477 subjects with a history of hypertension (Appendix A). Similarly, the estimates were similar after excluding 375 subjects who were taking hypertension or diabetes medications (Appendix A). In addition, PRs of substituting alternative sources stayed largely null when excluding 477 subjects with history of hypertension and 463 subjects with history of hypercholesterolemia separately. Moreover, the trends across the quintiles of the three main exposure variables were fairly similar after excluding 969 subjects who changed their long-term intake of fruits (Appendix A), and 876 subjects who reported that they had changed their long-term intake of vegetables (Appendix A). Finally, we did not find significant interactions between the total starchy vegetables, unhealthy starchy vegetables, or healthy starchy vegetable and age, sex, or BMI.

## 4. Discussion

The results from this cross-sectional study showed that a high intake of total starchy vegetables was not associated with a higher prevalence of MetS in a sample of Costa Rican adults. There were no significant associations between unhealthy starchy vegetables and healthy starchy vegetables and MetS when analyzed separately. Intake of potatoes, in particular, was not found to be associated with a higher MetS prevalence, as compared to other starchy vegetables. In contrast, it was observed that a higher intake of purple sweet potatoes was associated with a lower MetS prevalence, even though the trend did not reach statistical significance. When examining the MetS components individually, fasting blood glucose was lower with a higher intake of total starchy vegetables and healthy starchy vegetables, but not unhealthy starchy vegetables. 

Our findings suggest that a higher intake of starchy vegetables might not be contributing to the increased prevalence of MetS in this study population. Some studies found a positive association between a higher content of carbohydrates in vegetables and MetS; higher carbohydrate proportion in vegetables is associated with increased prevalence of developing MetS and hyperlipidemia in the United States (US) [18], as well as in some Asian countries, including China [19] and Korea [20]. However, we did not find an association of total, unhealthy, or healthy starchy vegetables with MetS or dyslipidemia in Costa Rican adults. As most starchy vegetables consumed in the US are French fries, potato chips, and mashed potatoes, the positive associations observed in the US, in part, might be due to dietary pattern being a confound. French fries, potato chips, and mashed potatoes are part of the Western dietary pattern, which is also high in red meats, processed meats, and refined flours [21,22]. In contrast, the consumption of starchy vegetables in Costa Rica is likely associated with a healthier dietary pattern predominantly relying on fresh vegetables, cereals, and legumes [23].

Consumption of potatoes, in particular, might not be contributing to the same metabolic responses and health effects in the Costa Rican sample, given the different varieties and cooking methods [24,25,26]. Potatoes popularly consumed in the US are white potatoes and Russet potatoes, which are primarily starch and with high glycemic index (GI) values [27]. These two types of potatoes are ideal for making French fries, given the rich content of starch and low moisture [27]. Prospective cohort studies conducted in the US showed that increased intake of fried potato products, including potato chips and fried potatoes, is strongly associated with a higher weight gain in women and men [22,28]. However, the native potato species commonly used in Costa Rica are cultivated in disparate conditions [29], which might contribute to different nutritional contents including total phenolics, carotenoid, anthocyanins, and vitamin C, and eventually result in distinguishable health outcomes [25]. Thus, the different varieties of commonly consumed potatoes or food preparation methods in Costa Rica might in part explain our null associations in this study.

Starchy root vegetables have a higher GI compared to non-starchy vegetables, and a higher GI can induce insulin resistance [30]. However, in a retrospective cohort study conducted in China, no significant associations were observed between medium or low GI groups and hyperlipidemia (*p* = 0.93) or MetS (*p* = 0.76) [19]. The GI of the starchy vegetables contained in our study belongs to the medium GI category (54–56 for starchy vegetables) [31]. Additionally, the GI for different varieties of potatoes showed a wide range from a medium of 56 for boiled red potatoes consumed cold to a high of 89 for boiled red potatoes [32]. Several varieties of potatoes commonly used in Costa Rica are imported from Canada and show intermediate GI values ranging from 59 to 70 [32]. Therefore, our findings of null associations are supported and consistent with the literature. 

Regardless of the high carbohydrate content, starchy vegetables are still rich sources of micronutrients and other healthy biological compounds [33]. These nutrients and biological compounds might play a role in the effect of starchy vegetables on macronutrients metabolism, antioxidant protection, and chronic disease status [27,33,34]. In rats, a potato-enriched diet significantly reduced plasma cholesterol and triglyceride levels, while improving the antioxidant status [33]. The antioxidant micronutrients, such as phenolic acid and carotenoids in potatoes scavenge superoxides and peroxyl radicals, and protect the lipoproteins from peroxidation, resulting in a protective effect of cardiovascular and metabolic status [34]. Cooked white potatoes indeed provide rich sources of potassium, magnesium, vitamin B-6, and ascorbic acid [27]. One study showed that baked and microwaved potatoes contain approximately twice the amount of ascorbic acid than boiled or fried potatoes [35]. Other starchy vegetables like purple sweet potatoes decrease the blood glucose levels in animal study [9], also leading to possible protective effects for diabetic patients. Many other potential compounds, such as phenolic compounds and saponins, might also be playing an important role in metabolic responses and human health [36]. 

Prior studies showed that reducing the consumption of starchy vegetables might result in improved insulin sensitivity [37]. In the Health Professionals Follow-up study and the Nurses’ Health Study, replacing starchy vegetables with one serving of non-starchy vegetable per day, decreases the risk of cardiovascular disease by approximately 20% [38]. In addition, starchy vegetables were shown to induce more insulin secretion and release than non-starchy vegetables, and by reducing the consumption of starchy vegetables, insulin sensitivity improved [37]. The substitution analysis indicated that there are no significant differences between starchy vegetables and non-starchy vegetables with respect to the prevalence of MetS. It is possible that people with a higher metabolic risk might switch to healthier diets that are richer in non-starchy vegetables and reverse causation could explain our null results. However, in sensitivity analysis, we excluded people with hypertension and hypercholesterolemia and people who changed their long-term intake of fruits or vegetables, and the results were still null. Further studies with longitudinal data are required to confirm our results.

To the best of our knowledge, this is the first study investigating the association between starchy vegetables intake and MetS in a Hispanic/Latino population. The strength of this study includes a population-based design in which the subjects were randomly selected from the target source population, who were representative within the matching strata of the original case-control study (i.e., sex, age, and area of residence). The results were also more likely to be generalizable to the entire Costa Rican population, given their lifestyle and socioeconomic characteristics. However, the study had few limitations. First, due to the observational study design, we cannot rule out residual confounds despite controlling for many potential confounders. Second, we cannot rule out reverse causation due to the cross-sectional nature of the study. Third, there is a potential for differential exposure misclassification, since a retrospective FFQ was used to determine the exposure starchy vegetable intake level. However, we believe that the risk of recall bias is minimal, because at the moment of the study, people in Costa Rica were not very aware of the associations between diet and MetS. Fourth, dietary assessment via an FFQ show more measurement errors than multiple 24-h foods recalls or diet records [39]. Nevertheless, because the FFQ was specifically developed and validated for the Costa Rican population and it is more appropriate for determining long-term food intake, we believe we still have a reliable assessment of the population intake. Lastly, there might be effect measure modification by sex in the association between starchy vegetables and metabolic responses. As our study population comprises more men than women, this potential heterogeneity could be masked in the whole population and the results could be mostly driven by the association among men.

In summary, our study showed that a higher intake of starchy vegetables is not associated with the prevalence of MetS and that starchy vegetables could be part of a healthy dietary pattern. Future studies are necessary to further investigate the potential cardiometabolic impacts of different types of starchy vegetables in other populations. 

## Figures and Tables

**Table 1 nutrients-13-01639-t001:** Characteristics of the study population by metabolic syndrome status in Costa Rican adults (*n* = 1881).

Variables	Metabolic Syndrome				*p* Value ^1^
	NO (*n* = 1123)		YES (*n* = 758)		
	Mean (or %)	SD	Mean (or %)	SD	
Age (year)	55.7	11.6	59.8	10.4	<0.0001
Women (%)	17.2%		36.3%		<0.0001
Urban residence (%)	41.2%		38.7%		0.2705
Income ($/month) ^2^	586	435	583	426	0.8129
BMI (kg/m^2^) ^3^	24.7	3.4	28.3	4.4	<0.0001
Waist/hip ratio	0.94	0.07	0.96	0.08	<0.0001
Current smoker (%)	26.2%		16.6%		<0.0001
Hypertension history (%)	13.8%		42.5%		<0.0001
High cholesterol history (%) ^2^	20.3%		31.0%		<0.0001
Alcohol (g/day)	6.81	14.99	5.47	12.30	0.0274
Systolic blood pressure (mm Hg)	129	21	144	20	<0.0001
Diastolic blood pressure (mm Hg)	79	10	86	11	<0.0001
HDL-C (mg/dL) ^2, 3^	42.3	9.3	38.6	6.8	<0.0001
LDL-C (mg/dL) ^2, 3^	131	36	124	36	0.0002
Fasting blood glucose (mg/dL) ^2^	72.4	13.6	81.5	28.8	<0.0001
Total triglycerides (mg/dL) ^2^	194	127	243	111	<0.0001
EE on daily activity (METs/day)	37.1	18.2	34.4	14.2	0.0148
Metabolic syndrome components (%)					
Abdominal obesity (waist circumference ≥102 cm in men or 88 cm in women)	3.4%		42.9%		<0.0001
Hypertriglyceridemia (triglycerides ≥150 mg/dL) ^2^	53.4%		92.6%		<0.0001
Low HDL cholesterol (high-density lipoprotein cholesterol <40 mg/dL in men or 50 mg/dlL in women) ^2^	46.9%		89.4%		<0.0001
High blood pressure (blood pressure ≥130 mm Hg systolic blood pressure or ≥85 mm Hg diastolic blood pressure)	46.1%		93.0%		<0.0001
High glucose (fasting glucose ≥100 mg/dL) ^2^	1.3%		12.7%		<0.0001
Dietary variables					
Total energy intake (kcal/day)	2483	713	2402	694	0.0120
Total fat (% of energy)	31.9	6.0	31.5	5.7	0.2950
Saturated fat (% of energy)	10.4	2.7	10.2	2.6	0.1011
Monounsaturated fat (% of energy)	12.0	4.1	11.7	3.6	0.1958
Polyunsaturated fat (% of energy)	6.1	2.0	6.2	2.0	0.2956
Trans fat (% of energy)	1.3	0.6	1.3	0.6	0.6455
Carbohydrate (% of energy)	55.3	7.5	55.9	7.3	0.0508
Protein (% of energy)	12.8	2.1	12.9	2.0	0.0894
Cholesterol (mg/kJ)	119.6	53.2	113.6	49.1	0.0114
Fiber (g/day)	22.9	6.0	23.6	5.8	0.0062
Total starchy vegetables (g/day)	70.1	45.5	76.1	49.2	0.0051
French fries	15.2	51.6	14.4	53.0	0.6183
Baked/boiled/mashed potatoes	18.3	24.1	21.5	26.7	0.0007
Potato chips	1.7	5.0	1.7	5.4	0.3868
Purple sweet potatoes	6.2	13.5	6.6	11.3	0.1734
Plantain	24.5	32.7	26.7	30.9	0.0549
Corn	2.3	6.0	2.7	5.7	0.0987
Cassava	16.2	18.8	19.0	23.4	0.0296

^1^ Chi-square tests for binary variables and two-sample *t*-tests and Wilcoxon rank-sum tests were used for normally distributed and non-normally distributed variables, respectively. ^2^ 126, 1, 25, 156, 3, 7, 23, 7, and 3 missing values, respectively. ^3^ BMI-Body Mass Index; HDL-C-High-density lipoprotein cholesterol; LDL-C-Low-density lipoprotein cholesterol.

**Table 2 nutrients-13-01639-t002:** Prevalence ratios of metabolic syndrome according to quintiles of total starchy vegetables, unhealthy starchy vegetables, and healthy starchy vegetables consumption (*n* = 1881).

	Quintiles					*p* for Trend
	1 (*n* = 376)	2 (*n* = 376)	3 (*n* = 377)	4 (*n* = 376)	5 (*n* = 376)	
Total starchy vegetables						
Median intake (g/day) ^1^	25.14 ± 12.95	45.54 ± 9.23	62.49 ± 8.71	85.20 ± 14.31	131.33 ± 43.15	
Crude model	1.0	1.00 (0.83, 1.20)	1.01 (0.84, 1.22)	1.09 (0.91, 1.31)	1.27 (1.07, 1.50)	0.0009
Adjusted model ^2^	1.0	0.98 (0.82, 1.16)	0.95 (0.80, 1.13)	0.92 (0.78, 1.09)	0.96 (0.82, 1.14)	0.6906
Fully adjusted ^3^	1.0	0.96 (0.81, 1.14)	0.94 (0.79, 1.12)	0.92 (0.78, 1.09)	0.99 (0.84, 1.17)	0.8128
Unhealthy starchy vegetables						
Median intake (g/day) ^1^	2.74 ± 5.10	10.98 ± 3.74	20.22 ± 5.56	33.74 ± 9.16	68.69 ± 42.54	
Crude model	1.0	0.84 (0.69, 1.02)	1.13 (0.95, 1.34)	1.07 (0.90, 1.27)	1.15 (0.97, 1.36)	0.0119
Adjusted model ^2^	1.0	0.82 (0.68, 0.99)	1.03 (0.87, 1.21)	0.98 (0.83, 1.15)	0.93 (0.79, 1.09)	0.8223
Fully adjusted ^3^	1.0	0.86 (0.71, 1.03)	1.06 (0.90, 1.24)	0.95 (0.80, 1.12)	0.97 (0.82, 1.14)	0.9632
Healthy starchy vegetables						
Median intake (g/day) ^1^	11.76 ± 7.63	24.66 ± 6.24	38.40 ± 7.56	54.70 ± 9.70	86.59 ± 39.90	
Crude model	1.0	0.99 (0.83, 1.19)	0.93 (0.77, 1.12)	1.00 (0.84, 1.19)	1.19 (1.01, 1.41)	0.0134
Adjusted model ^2^	1.0	0.96 (0.82, 1.14)	0.88 (0.74, 1.05)	0.87 (0.74, 1.03)	0.92 (0.78, 1.07)	0.3094
Fully adjusted ^3^	1.0	0.97 (0.83, 1.13)	0.88 (0.75, 1.02)	0.85 (0.73, 0.99)	0.93 (0.80, 1.07)	0.4098

^1^ Medians ± Interquartile ranges (IQRs). ^2^ Adjusted for age, sex, current smoking status, urban residence, and income. ^3^ Additionally adjusted for history of hypertension, alcohol, and total energy expenditure.

**Table 3 nutrients-13-01639-t003:** Least square means of metabolic syndrome components according to quintiles of total starchy vegetables, unhealthy starchy vegetables, and healthy vegetables consumptions in crude and fully adjusted models (*n* = 1881).

		Quintiles					*p* for Trend
		1 (*n* = 376)	2 (*n* = 376)	3 (*n* = 377)	4 (*n* = 376)	5 (*n* = 376)	
Total starchy vegetables							
Median intake (g/day)		25.1 ± 13.0	45.5 ± 9.2	62.5 ± 8.7	85.2 ± 14.3	131.3 ± 43.2	
Adjusted Waist circumference (cm) ^1^	Crude model	104.7	104.3	104.2	103.8	102.6	<0.0001
	Fully adjusted	104.0	103.7	104.1	104.3	103.6	0.4247
Triglycerides (mg/dL) ^2, 3^	Crude model	213	218	215	211	211	0.5618
	Fully adjusted	210	215	215	212	216	0.6586
HDL cholesterol (mg/dL) ^2^	Crude model	40.5	40.3	40.3	41.0	42.1	0.0026
	Fully adjusted	40.7	40.7	40.6	40.8	41.5	0.1661
Systolic blood pressure (mm Hg)	Crude model	134	133	134	136	140	<0.0001
	Fully adjusted	136	134	135	135	137	0.3448
Diastolic blood pressure (mm Hg)	Crude model	82	81	82	81	82	0.4838
	Fully adjusted	82	81	82	81	82	0.6109
Fasting blood glucose (mg/dL)	Crude model	79.6	75.5	76.6	74.1	74.7	0.0040
	Fully adjusted	79.8	75.4	76.6	73.9	74.7	0.0040
Unhealthy starchy vegetables							
Median intake (g/day)		2.7 ± 5.1	11.0 ± 3.7	20.2 ± 5.6	33.7 ± 9.2	68.6 ± 42.5	
Adjusted Waist circumference (cm) ^1^	Crude model	104.5	104.3	104.0	104.1	102.7	<0.0001
	Fully adjusted	103.9	103.9	104.1	104.1	103.6	0.1968
Triglycerides (mg/dL) ^2^	Crude model	211	216	228	210	203	0.1158
	Fully adjusted	210	214	228	210	206	0.2410
HDL cholesterol (mg/dL) ^2, 3^	Crude model	39.9	40.8	40.7	40.9	42.0	0.0014
	Fully adjusted	40.2	41.0	40.6	41.0	41.5	0.0508
Systolic blood pressure (mm Hg)	Crude model	135	133	136	136	137	0.0590
	Fully adjusted	135	135	136	135	135	0.5030
Diastolic blood pressure (mm Hg)	Crude model	82	82	82	81	82	0.3634
	Fully adjusted	82	82	82	81	82	0.4423
Fasting blood glucose (mg/dL)	Crude model	77.3	77.0	76.1	74.6	75.5	0.1703
	Fully adjusted	77.4	77.0	76.1	74.4	75.5	0.1635
Healthy starchy vegetables							
Median intake (g/day)		11.8 ± 7.6	24.7 ± 6.2	38.4 ± 7.6	54.7 ± 9.7	86.6 ± 39.9	
Adjusted Waist circumference (cm) ^1^	Crude model	104.6	104.8	103.8	103.7	102.7	<0.0001
	Fully adjusted	103.8	104.1	103.9	104.1	103.7	0.5396
Triglycerides (mg/dL) ^2^	Crude model	218	206	215	211	217	0.7998
	Fully adjusted	212	205	215	213	223	0.1061
HDL cholesterol (mg/dL) ^2, 3^	Crude model	40.9	40.5	40.0	40.3	42.6	0.0022
	Fully adjusted	41.0	40.9	40.1	40.3	42.0	0.0810
Systolic blood pressure (mm Hg)	Crude model	133	135	134	135	139	0.0002
	Fully adjusted	136	135	135	134	136	0.7570
Diastolic blood pressure (mm Hg)	Crude model	82	82	81	82	81	0.4671
	Fully adjusted	82	82	82	82	82	0.6801
Fasting blood glucose (mg/dL)	Crude model	77.7	77.6	74.9	75.8	74.3	0.0150
	Fully adjusted	77.8	77.4	75.1	75.8	74.3	0.0236

^1^ Waist circumference is adjusted for age and BMI. ^2^ 7, 25 missing observations. ^3^ HDL cholesterol-High-density lipoprotein Cholesterol.

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
