# Peer review of "Starchy Vegetables and Metabolic Syndrome in Costa Rica"

_nutrients, 2021, doi:10.3390/nu13051639_

Round 1

Reviewer 1 Report

I read with interest the manuscript entitled 'Starchy Vegetables and Metabolic Syndrome in Costa Rica' in which the authors evaulated the association of starchy vegetables with the prevalence of Metabolic Syndrome. I would propose minor corrections: 

1) I would prefer the term Οdds Ratio instead of Prevalence Ratio

2) The associations in Tables 2 and 3 should be adjusted for the variables differing between 2 groups in Table 1 (ie. sex, age, smoking, cholesterol, EE, Total energy intake, fibers). Parameters included in the definition of Metabolic Syndrome (such as TG, HDL, etc) should not be used as confounding factors. 

3) The authors reported that substituting non-starchy vegetables for the three types of starchy vegetables were not associated with a significantly different prevalence of MetS. If i am not wrong, the subjects did not subitute non-starchy for starchy vegetables during the study. The corresponding risk refers to the difference between 2 groups. Therefore, i would rephrase the corresponding results into: There was no difference regarding the risk of Metabolic Syndrome between total starch and non-starchy vegetables etc

Author Response

We appreciate the insightful comments from the reviewers. We have answered below to each of their questions and we have highlighted in yellow changes made to the manuscript.  

Review 1:

I would prefer the term Οdds Ratio instead of Prevalence Ratio.

Authors reply: We apologize for not making clear enough that we did not conduct logistic regression but a log-binomial regression. Therefore, our measures of association are not odds ratios, but prevalence ratios. We have highlighted that section in the article now. Although choosing to report odds ratios vs prevalence ratios is highly discussed in the literature, we chose to run log-binomial models rather than logistic models because metabolic syndrome is highly prevalent in this population. That way we can avoid the potential of odds ratios to “overestimate” the associations (Stat Med. 2016 Dec 30; 35(30): 5730–573).

The associations in Tables 2 and 3 should be adjusted for the variables differing between 2 groups in Table 1 (ie. sex, age, smoking, cholesterol, EE, Total energy intake, fibers). Parameters included in the definition of Metabolic Syndrome (such as TG, HDL, etc) should not be used as confounding factors.

Authors reply: We agree with the reviewer and we did not adjust for parameters included in the definition of Metabolic Syndrome. Confounding variables were selected based on prior knowledge and the association of these covariates with the exposure among people without MetS from supplementary table 1. We want to emphasize that including the components of the metabolic syndrome in Table 1 was done only for descriptive purposes, since we did not adjust for any of them in the models. Not all the covariates included in table 1 were included in the final models because they did not seem to impact the effect estimates.

The authors reported that substituting non-starchy vegetables for the three types of starchy vegetables were not associated with a significantly different prevalence of MetS. If I am not wrong, the subjects did not substitute non-starchy for starchy vegetables during the study. The corresponding risk refers to the difference between 2 groups. Therefore, I would rephrase the corresponding results into: There was no difference regarding the risk of Metabolic Syndrome between total starch and non-starchy vegetables etc.

Authors reply: We agree with the reviewer that the subjects did not make any substitution. We have reviewed the text properly to clarify that this is a “theoretical substitution”. These types of models are commonly referred as “substitution models” in the literature and that is why we used that language, but we agree with the reviewer that the substitution is not real. In line 157 of the methods section we have added “Theoretical substitutions were assessed with substitution models” to emphasize that these substitutions are not real. Furthermore, revisions have been made in the discussion, line 359. This reads now as “The substitution analysis indicated there are no significant differences between starchy vegetables and non-starchy vegetables with respect to the prevalence of MetS”

Reviewer 2 Report

This study describes the association between starchy vegetables consumption and MetS in a population-based sample of Costa Rican adults. The main finding is that while consumption of starchy vegetables is not associated with a higher prevalence of MetS in Costa Rica, the consumption of total starchy and healthy starchy vegetables consumption is negatively associated with fasting blood glucose.

The study is well-executed, the manuscript is mostly well-written language-wise, and the results are of interest to the readers of this Journal.

My point problem with the manuscript is how the results are presented. The authors claim that higher intakes of starchy vegetables were not associated with a higher prevalence of MetS in the Abstract. However, I think it would be more appropriate to present the main finding more in line as in the Results: “The crude models showed significant higher prevalences of MetS across quintiles of starchy vegetables. However, after adjusting for confounders, no significant trends were observed”. Thus, there were significantly higher prevalence of MetS which disappeared after adjustments.

Another problem is the sentence in the Abstract “Healthy starchy vegetables showed a non-significant protective trend for MetS, driven by purple sweet potatoes (PRhighest quintile 24 vs. 1st: 0.88; 95% CI: 0.75, 1.04; P for trend: 0.2853).” However, there was not any protective trend for healthy starchy vegetables regarding Mets as in Table 2 the p-value for this “trend” is 0.4098. And it is debatable if a trend with p-value of 0.2853 for purple sweet potatoes indeed is a trend. The finding on purple sweet potatoes is perhaps interesting regionally in Central America but it is highlighted undeservedly strongly.

Thus, the Abstract should be written to line up with the results.

Furthermore, Discussion on study limitations: although it is true that the subjects “are representative within matching strata of the original case-control study” they fail to highlight that there is quite a small proportion of women in this cohort (<30%). It should be discussed how this is reflected in the results.

Some more minor points:

In Abstract: it is mentioned that the cohort is “large”. It really is not that large nowadays; I would omit this word throughout the manuscript.

Page 2: second line. I don’t think reference 9 says anything about blood glucose elevation rates as it is a molecular analysis of potatoes. And the same problematic referencing on ref 9 occurs in the Discussion

Page 2: his circumference, should be hip circumference

Page 3, first lines. There are no mention of boiled potatoes. Perhaps the authors should mention in the manuscript if this type of cooking method is not at all practiced in Costa Rica.

Page 3: this is cumbersome ”We excluded subjects who was not possible”

Page 3: ”two-samples t-tests” should be ”two-sample t-tests”

Tables: there are too many decimals in the numbers throughout the Tables.

Page 9: the discussion on GI should also mention this recent paper: Jenkins et al. Glycemic Index, Glycemic Load, and Cardiovascular Disease and Mortality. N Engl J Med. 2021

Reference 34: gives the name of the journal twice.

Author Response

We appreciate the insightful comments from the reviewers. We have answered below to each of their questions and we have highlighted in yellow changes made to the manuscript.  

Reviewer 2:

My point problem with the manuscript is how the results are presented. The authors claim that higher intakes of starchy vegetables were not associated with a higher prevalence of MetS in the Abstract. However, I think it would be more appropriate to present the main finding more in line as in the Results: “The crude models showed significant higher prevalences of MetS across quintiles of starchy vegetables. However, after adjusting for confounders, no significant trends were observed”. Thus, there were significantly higher prevalence of MetS which disappeared after adjustments.

Authors reply: Revisions have been made accordingly in the abstract, line 22. It reads now: “Higher intakes of starchy vegetables were associated with a higher prevalence of MetS in crude models, but no significant trends were observed after adjusting for confounders.” This statement is now lined up with the results.

Another problem is the sentence in the Abstract “Healthy starchy vegetables showed a non-significant protective trend for MetS, driven by purple sweet potatoes (PRhighest quintile 24 vs. 1st: 0.88; 95% CI: 0.75, 1.04; P for trend: 0.2853).” However, there was not any protective trend for healthy starchy vegetables regarding Mets as in Table 2 the p-value for this “trend” is 0.4098. And it is debatable if a trend with p-value of 0.2853 for purple sweet potatoes indeed is a trend. The finding on purple sweet potatoes is perhaps interesting regionally in Central America but it is highlighted undeservedly strongly.

Authors reply: We agree with the reviewer and we have deleted that from the abstract. 

Furthermore, Discussion on study limitations: although it is true that the subjects “are representative within matching strata of the original case-control study” they fail to highlight that there is quite a small proportion of women in this cohort (<30%). It should be discussed how this is reflected in the results.

Authors reply: We agree with the reviewer that this is a limitation of the study and we have added revisions to the discussion section, line 329. It reads as follows: “Lastly, there may be effect measure modification by sex in the association between starchy vegetables and metabolic responses. Because our study population comprises more men than women, this potential heterogeneity could be masked in the whole population and the results would be mostly driven by the association among men.

Some more minor points:

In Abstract: it is mentioned that the cohort is “large”. It really is not that large nowadays; I would omit this word throughout the manuscript.

  • Authors reply: We have deleted large from the abstract and throughout the manuscript.

Page 2: second line. I don’t think reference 9 says anything about blood glucose elevation rates as it is a molecular analysis of potatoes. And the same problematic referencing on ref 9 occurs in the Discussion

  • Authors reply: We have reviewed that part accordingly and added one more reference, line 46. It reads now: “… and purple sweet potatoes has been shown to decrease blood glucose levels in a diabetic rats study because of the antioxidant characteristics of their flavonoid content [9]. Moreover, the molecular characteristic of amylose-to-amylopectin ratio of sweet potatoes [10] might also contribute to variations on starch contents with respect to metabolic responses comparing to other starchy vegetables”.

Page 2: his circumference, should be hip circumference

  • Authors reply: Thank you for catching that typo. We have corrected it.

Page 3, first lines. There are no mention of boiled potatoes. Perhaps the authors should mention in the manuscript if this type of cooking method is not at all practiced in Costa Rica.

  • Authors reply: We apologize for the confusion. The FFQ asked about baked, boiled and mashed potatoes together, so every time we referred to baked potatoes we were actually referring to the three items together. We have modified that in the text accordingly.

Page 3: this is cumbersome ”We excluded subjects who was not possible”

  • Authors reply: We have rephrased that paragraph. Line 114, it reads now “1) if it was not possible to classify the presence or absence of MetS due to missing data on more than two MetS components (n=29), 2) subjects who had missing data on major potential confounders (current smoking status, history of hypertension, and BMI; n=21), 3) subjects who had caloric intakes greater than 5,000 kcals or less than 500 kcals (n=19), and 4) subjects who had history of diabetes (as those with diabetes were likely to change their starchy vegetable consumption; n=324).”

Page 3: ”two-samples t-tests” should be ”two-sample t-tests”

  • Authors reply: Thank you for catching that typo. We have corrected it.

Tables: there are too many decimals in the numbers throughout the Tables.

  • Authors reply: We agree with the reviewer and we have rounded figures accordingly.

Page 9: the discussion on GI should also mention this recent paper: Jenkins et al. Glycemic Index, Glycemic Load, and Cardiovascular Disease and Mortality. N Engl J Med. 2021

  • Authors reply: We have added that reference to the manuscript.

Reference 34: gives the name of the journal twice.

  • Authors reply: We have corrected that. This is now reference 36, since two more references were added to the manuscript in this revision.